# Analysis of Food Perception in Slim, Overweight, or Obese Individuals

**DOI:** 10.3390/nu17132054

**Published:** 2025-06-20

**Authors:** Anna Celina Durma, Maja Sosnowska, Adam Daniel Durma, Adam Śmiałowski, Leszek Czupryniak

**Affiliations:** 1Department of Diabetology and Internal Medicine, Medical University of Warsaw, 02-097 Warsaw, Poland; 2Department of Endocrinology and Radioisotope Therapy, Military Institute of Medicine—National Research Institute, 04-141 Warsaw, Poland; 3Department of Endocrinology and Metabolic Diseases, Polish Mother’s Memorial Hospital, Research Institute, Medical University of Lodz, 93-338 Lodz, Poland

**Keywords:** eating habits, estimation of calories, obesity, obesity cause, portion size

## Abstract

Introduction: Obesity is a systemic disease leading to many complications. One of the causes of obesity is excessive energy intake in relation to its expenditure. Assessing portion sizes and estimating caloric intake is crucial in maintaining a healthy body weight and combating obesity. Objectives: To evaluate the impact of BMI on the perception of portion sizes and their estimated caloric content. Patients and methods: The anonymous survey was filled out by 205 patients. The survey contained questions regarding different meals. Pictures of main meals were presented and individually assessed by the study participants. Next, they were divided into groups, individuals with normal weight (BMI < 25 kg/m^2^), who were overweight (BMI 25–29.9 kg/m^2^), and with obesity (BMI ≥ 30 kg/m^2^), to analyze the differences in food perception and caloric estimation. Results: The study did not demonstrate significant differences in the subgroups’ estimated portion sizes of most main meals. No statistical significance was found in the estimated caloric content of the indicated main meal portions across the studied groups. Obese, overweight, and normal-weighted patients assess food portion size similarly. Conclusions: BMI has no significant impact on caloric estimation. The majority of the population estimate the caloric value of the meals inappropriately. Nevertheless, patients with excessive body weight (overweight and obesity) have a tendency (but not statistically significant) to underestimate the caloric value of full meals compared with people with normal BMI. Incorrect calorie estimation may lead to consuming bigger meal portions in patients with overweight and obesity.

## 1. Introduction

Obesity is a lifestyle disease associated with abnormal body weight, affecting over a billion people worldwide. According to the World Health Organization (WHO) data from 2022, 1 in 8 people globally is obese [1]. International guidelines diagnose obesity based on the Body Mass Index (BMI), calculated as the quotient of body weight and height squared [2]. Obesity is defined as a BMI ≥ 30 kg/m^2^, while overweight is defined as a BMI of 25.0–29.9 kg/m^2^. Normal body weight corresponds to a BMI of 18.5–24.9 kg/m^2^.

The causes of obesity include environmental factors, genetic predisposition, medications, coexisting diseases, and an unhealthy lifestyle. According to Marc Lalonde’s concept, lifestyle remains the critical factor influencing health, encompassing physical activity, diet, substance use, and stress management [3]. Improper dietary patterns, often established in childhood, significantly contribute to the development of obesity during adolescence and adulthood [4]. The primary cause of unhealthy eating habits is usually insufficient education regarding healthy diet, the caloric content of meals, and incorrect eating patterns [5]. Portion size is another contributing factor to the development of obesity. Moreover, over recent years, meal portion sizes have been systematically increasing, which may have significantly influenced the increasing prevalence of obesity [6].

The systematic increase in the proportion of people with abnormal body weight or eating disorders, coupled with the worldwide availability of highly processed, high-calorie foods, can exacerbate distortions in portion size perception and subsequently contribute to further weight gain in obese individuals.

To reduce body weight, it is essential to assess the quantity and quality of meals consumed. The evaluation should include carbohydrate intake, such as fiber or fat content. Low-fat diets (containing <30% of daily calories from fat) facilitate weight reduction. The effectiveness of low-fat diets is multifactorial. One reason for weight reduction during a low-fat diet is the decreased energy value of such meals [7]. Also, low-carb diets have been shown to help decrease body weight in obese persons [8].

Considering the above, we conducted this study to determine whether a relationship exists between body weight and the ability to assess main meal sizes adequately. Several studies have shown that people with an increased BMI tend to choose larger portions [9,10]. This fact can help to understand one of the causes of obesity. On the other hand, other studies found that there were no significant differences in chosen portion size of meals between people with normal and excessive body weight [11,12,13]. Thus, the effect of BMI on the choice of portion size still seems unclear and requires more research. We also aimed to test the hypothesis that obese people are more likely to underestimate meal caloric value than non-obese individuals, which—if confirmed—could help understand the phenomenon of obese people becoming more obese over time (despite all the efforts made to lose weight). It is proved that people who reported less knowledge about calories were more likely to be overweight and obese [14]. It is therefore worth examining and assessing the direction of calorie estimation by people with an increased BMI.

## 2. Materials and Methods

### 2.1. Questionnaire Development

The anonymous questionnaire was designed by authors based on their experience and competence in obesity and nutritional disorders treatment. It was then shared on several popular websites devoted to healthy nutrition and weight management or on social media. Each adult person who accessed these websites was welcome to fill it in. We also encouraged patients in outpatient clinics to fill out the forms. As we aimed to gather data across the whole spectrum of body weight, we decided not to limit access to the questionnaire, with one exception: all participants had to be ≥18 years old. Accordingly, weight-related disorders (e.g., diabetes, hypertension, etc.), education level, and social status were not excluded from the study. Participation in the study was anonymous and, in the majority, remote (the questionnaire was filled out online). All data on an individual’s health and anthropometric values were self-declared. Participants had unlimited time to answer all the questions. However, the minimum time required to complete the questionnaire was estimated to be 35 min.

The questionnaire was designed and approved by a national expert in the field of diabetology and obesitology and at the same time the supervisor of the study (LC). The survey questions were designed to answer the hypothesis regarding the influence of BMI on the perception of meals. Moreover, some questions were added to assess participants’ ability to evaluate the caloric value of meals. A similar survey was developed for other studies that investigated a similar hypothesis [15,16]. It consisted of two parts. The first part consisted of questions about the patient’s general data and eating habits. The second part consisted of 10 color photos of main meals, which were prepared and photographed exclusively for this study by one of the authors (MS). The meals were selected based on their worldwide acceptance (i.e., they are known and consumed regardless of local food culture), with one exception being a typical Polish main course (minced meat cutlet with potatoes and cucumber salad). Each photograph was accompanied by a detailed description of the meal components; for instance, the addition of olive oil, which might not be more noticeable in the pictures, was mentioned in the meal description. Standard-sized cutlery was placed next to the plates (of uniform size for all meals) in the photographs to help participants better assess portion sizes (Figure 1). Thanks to the use of standard sizes of plates and cutlery, it was possible to obtain standardized answers. Thanks to the uniformed framing of the photos of meals and the uniformed background of the photos, the risk of error in the subjective assessment of the size of the portion was minimized. Most validated food atlases use similar principles to depict portion sizes as do the placement of food items in photographs and commonly used household kitchen utensils, such as cutlery and tableware [17].

To assess portion size perception in relation to BMI, each participant was asked to subjectively determine the portion size of each meal by providing an ordinal value from 1 to 10, where 1 indicated a very small meal, and 10 indicated a considerable meal. We also investigated the satiety level after eating the pictured portion to see differences between the study groups. Volunteers also assessed whether they would be satiated with the given portion (yes/no) and how long it would take before they felt hungry again (in hours). The degree of satiety after consuming the meal was rated on a scale of 1–10, where 1 meant still hungry and 10 meant fully satiated. The estimated time until feeling hungry again was indicated by choosing one of the following options: within an hour, 1–2 h, 2–3 h, or more than 3 h. Finally, respondents were asked to estimate the meal’s caloric content by entering a number representing the absolute caloric value of the meal to the best of their knowledge.

The evaluated meals included typical breakfast, lunch, and dinner items (10 meals in total):

Breakfast meals:Oatmeal with yogurt.Fried eggs in olive oil with Parma ham and avocado.Whole grain bread sandwiches with cottage cheese, bell peppers, and tomatoes.

Lunch/dinner meals:
Hamburger with fries.Sushi.Cutlet with potatoes and salad.Greek salad.Pizza with salami and pepperoni.Brown pasta with pesto, tomatoes, and mozzarella.Tomato cream soup.

The questionnaire and meal photographs are included in this work as Appendix A.

### 2.2. Study Group

The completed questionnaires were collected from 205 volunteers aged 18 to 77 years. The body weight of the participants ranged from 44 kg to 170 kg. The detailed study group characteristics are presented in Table 1. The study group was divided into subgroups depending on BMI. At the time of the study, the average age of the participants was 39.7 ± 13.1 years. The average BMI of the participants was 27 kg/m^2^ (±6.3 kg/m^2^), with approximately 1/4 of the participants being obese (24.4%), 30.7% being overweight, 40.0% having a normal weight, and 4.9% being underweight. Among the participants, 69.3% had higher education, 30.2% had secondary education, and less than 1% had primary education.

The study group was divided into three subgroups according to BMI: the standard weight group (BMI < 24.9 kg/m^2^, *n* = 92), the overweight group (BMI 25–29.9 kg/m^2^, *n* = 66), and the obese group (BMI ≥ 30 kg/m^2^, *n* = 47). Differences in responses to questions regarding the assessment of portion size, degree of satiety after consuming the meal, the estimated caloric content of the meals, evaluation of whether the meal is healthy, and whether the meal would maintain satiety for at least 2 h were analyzed.

### 2.3. Statistical Analysis

Statistical analysis was performed using SPSS Statistics (IBM v23). The Kolmogorov–Smirnov test was conducted to verify the normality of the data distribution. Quantitative variables were presented as mean (M) and standard deviation (SD) for normally distributed data or as medians (Med.) and interquartile ranges (IQR) for non-normally distributed data, as well as for describing ordinal variables. Qualitative (nominal) variables were analyzed using the Chi-square and Fisher’s Exact tests. For ordinal and quantitative data, the differences between the study subgroups were analyzed using the Kruskal–Wallis test. Post hoc tests were performed when statistical significance was found (Bonferroni). The odds ratio for overestimating meal portions was calculated by dividing the study group into individuals with excessive body weight (BMI ≥ 25 kg/m^2^) and those with normal body weight (BMI < 24.9 kg/m^2^). Results with *p* < 0.05 were considered statistically significant.

## 3. Results

### 3.1. General Meals Perception

No statistical significance was found in any of the examined variables and aspects for the hamburger with fries; whole grain bread sandwiches with cottage cheese, bell peppers, and tomatoes; sushi; cutlet with potatoes and salad; Greek salad; pizza with salami and pepperoni; tomato cream soup; and fried eggs in olive oil with Parma ham and avocado. The detailed results of the study group are presented in Table 2 and Table 3.

### 3.2. Portion Size Assessment

The portion size assessment for the whole grain pasta with pesto, mozzarella, and tomatoes was statistically significant. Each group’s median portion size rating was 8 (on a scale of 1–10). Statistical significance was observed between individuals with normal weight and those with obesity. Other than that, there were no differences in the portion size assessment for other dishes.

### 3.3. Feeling of Fullness

Statistical significance was noted regarding the degree of fullness after consuming the presented portion of the pasta meal. In the groups with BMI < 24.9 kg/m^2^ and BMI 25–29.9 kg/m^2^, respondents rated their fullness at 9/10. For the obese individuals, the median fullness rating was 8/10. The difference in the fullness assessment was statistically significant (*p* = 0.006) between the groups with BMI < 24.9 kg/m^2^ and BMI ≥ 30 kg/m^2^. A similar difference was observed in the case of the oatmeal with natural yogurt meal. In the group with BMI < 24.9 kg/m^2^, 5 out of 92 individuals responded that they would feel complete with such a meal. Among persons with a BMI of 25–29.9 kg/m^2^, 8 out of 63 individuals responded affirmatively, and in the BMI ≥ 30 kg/m^2^ group, 9 out of 50 individuals responded affirmatively. The feeling of fullness was statistically significantly more often observed in the group with BMI < 24.9 kg/m^2^ than with BMI ≥ 30 kg/m^2^ (*p* = 0.003). At the same time, no difference was recorded regarding the degree of fullness in these groups.

### 3.4. Estimated vs. Real Calorie Value

Less than a quarter of the participants in the group estimated the caloric content of meals correctly. The highest percentage of correct answers indicating the actual caloric content of the meal was 22.4%, and it pertained to the sushi meal. The lowest percentage of correct answers was for the oatmeal with yogurt meal, at 8.8%. The results are presented in Table 4.

Individuals with excessive body weight (BMI ≥ 25 kg/m^2^) similarly assess the caloric value compared with those with normal body weight. There might be a tendency to underestimate meal calories in the obese and overweight subgroups; nevertheless, the results are still not statistically significant. No statistical significance was found in the assessment of the risk of overestimating the caloric content of main meals by individuals with an abnormal body weight, except for the whole grain pasta with tomatoes [OR 0.51 (CI 0.95, 0.29–0.89), *p* = 0.022]. The detailed results are presented in Table 5.

### 3.5. Self-Assessment of Diet

In the first part of the survey, respondents answered questions about their diet habits. One of them was a self-assessment of their way of nutrition. Overall, 53.7% of respondents answered that they eat properly, 36.1% stated that they eat improperly, and 10.2% of respondents answered that they do not pay attention to their diet.

## 4. Discussion

The data for this study were collected using a specially designed questionnaire, developed based on the authors’ long-term experience in obesity management and nutritional intervention in various diseases. Since food behavior is closely related to local customs and culinary diversity, we were unable to identify a suitable existing questionnaire; thus, we took great care in developing one specifically for this study, following general recommendations for data collection in metabolic research.

In our study, no differences were observed in the estimation of calorie content for almost all meals among patients with a BMI indicating a normal weight, overweight, or obesity. Similarly, no significant differences were found in the assessment of portion sizes or the degree of satiety for most meals across the studied groups. Consequently, no differences were noted in the reported calorie values of the selected dishes.

Furthermore, our results indicate that only a minority of respondents accurately estimated the calorie content of meals. Among all participants, the highest percentage of correct calorie estimations was for the sushi meal, at 22.4%, meaning fewer than one-fourth of respondents answered correctly. However, no statistically significant differences in calorie estimation accuracy were found between the study groups.

In our study, the calorie estimates reported by obese individuals tended to be slightly higher in some cases compared with other groups. For example, the median estimated calorie content for a sushi meal was approximately 450 kcal for patients with a normal weight, about 475 kcal for those with a BMI of 25–29.9 kg/m^2^, and 500 kcal for individuals with obesity (BMI ≥ 30 kg/m^2^). Similarly, higher calorie values were reported by individuals with obesity for hamburgers with fries. This may suggest that people with obesity tend to overestimate the caloric content of these meals. A similar conclusion was reached by Block et al., who found that adults with a higher BMI tended to estimate meal calorie content to be higher [18].

For dishes such as whole grain pasta with mozzarella and tomatoes, Greek salad, and whole grain sandwiches with various fillings, respondents with a BMI ≥ 30 kg/m^2^ provided somewhat lower calorie estimates. In these cases, the meals were predominantly perceived as healthy, which may explain the underestimation of their caloric content. Carels et al. also found that respondents tend to underestimate the calorie content of foods they perceive as healthy [19].

Additionally, when comparing groups with normal and excessive body weight (BMI < 25 kg/m^2^ vs. BMI ≥ 25 kg/m^2^), we observed a trend of caloric underestimation for main meals among individuals with excessive body weight, with the exception of the cutlet with a potato dish. This may help explain why individuals with excessive body weight may consume more food.

However, overall, the median estimated calories for most meals were similar across all study groups.

One statistically significant finding in our study was that respondents who evaluated how full they would feel after consuming a portion of whole grain pasta indicated a lower satiety level compared with other groups. This suggests that they might be inclined to consume larger portions of the same meal. This could contribute to the development of obesity, as individuals with lower satiety may require larger portion sizes to feel full compared with those with a BMI < 30 kg/m^2^. Although the differences in responses were minimal, they appear to be clinically significant. Reduced satiety may result in increased food consumption and calorie intake. It is known that a daily energy surplus of 200 kcal can lead to a weight gain of 10 kg over a year. Even small differences in satiety can contribute to gradual weight gain.

Our study showed that portion sizes were assessed similarly regardless of patients’ BMI. We also found that the respondents frequently misestimated the caloric value of meals. More than half of the respondents reported eating properly, while about 10% admitted to not caring about their diet. At the same time, only 44.9% of study participants had a normal body weight. This indicates that some respondents were unable to accurately assess the quality of their diet. This suggests that one factor contributing to being overweight and having obesity is the consumption of poor-quality meals—high-calorie foods that are often available in relatively small portions.

No statistically significant differences in calorie estimation accuracy were found between the study groups. This suggests that inaccurate calorie estimation may not be a key mechanism in obesity development. Therefore, educating the public on calorie estimation may not be a critical strategy in combating obesity, especially since it is a difficult skill to acquire. Among our participants, nearly 70% reported a higher education level, yet in most cases, their calorie estimates were incorrect. In the United States, the government has mandated calorie labeling on fast food menus to improve consumers’ calorie awareness. However, research by Harnack et al. did not demonstrate that food labeling significantly reduced calorie intake [20]. Improved calorie estimation skills did not necessarily lead to weight loss. Greater effects were observed when product labels included recommendations on daily calorie intake [21]. On the other hand, calorie estimation may still play a role in weight management, and it has been shown that calorie counting can promote weight loss [22]. More substantial weight loss has been observed among individuals who frequently sought dietary advice [22]. Studies of fast food consumers found that patrons often underestimate the calories they consume [18], which may contribute to overeating and obesity. Carels et al. also showed that individuals with higher BMIs had greater inaccuracies in estimating calorie content [23].

Our study found that the median estimated calories for most meals were similar across study groups. One possible explanation is that perceptions of calorie content depend more on the visual size of the food than its nutritional quality. In our study, all meals were presented using the same tableware, controlling for the visual size effect and allowing a focus on the food’s content. Since the portion sizes appeared similar relative to the plate, this may have contributed to the similar responses across the groups. This phenomenon is known as the Delboeuf effect, which suggests that the same portion appears smaller or larger depending on the size of the plate [24].

Some studies support our finding—showing no correlation between BMI and preferred portion sizes [11,12]. In contrast, other studies have shown that individuals with higher BMIs tend to choose larger food portions [9,10].

The number of calories consumed is also influenced by the portion sizes served [25,26]. Wansink et al. investigated how portion size affects snack consumption using popcorn served in medium and large containers. Popcorn consumption was 45.3% higher when served in larger containers. Interestingly, the study also assessed the consumption of less palatable snacks, including stale popcorn, and still found a 33.6% increase in consumption from larger containers [27]. Similar results were reported by Diliberti et al., who found that participants consumed about 43% more calories when the main course (pasta) portion was 50% larger than standard [28].

Recently, increased food availability—especially processed, highly palatable foods—has been observed in both developed and developing countries [29]. These products often contain high levels of saturated fats and simple carbohydrates. Research by Lampure et al. found that preferences for high-fat foods were associated with an increased risk of obesity [30]. Bartoshuk et al. also noted that obese individuals, compared with non-obese individuals, had a greater preference for sweet and high-fat foods [31]. Therefore, obesity is influenced not only by the quantity of food consumed but also by its caloric density and quality.

The causes of obesity should be directly attributed to excessive caloric intake relative to expenditure. Assessing the factors leading to this imbalance allows progress in treatment methods and management strategies for individuals with excessive body weight. Improving the quality of eating habits appears to be one of the key strategies for addressing the problem of obesity. Other potential approaches include controlling portion sizes, enhancing individuals’ ability to estimate caloric content accurately, and developing better awareness of personal calorie requirements.

In summary, our study demonstrated that portion sizes are assessed similarly across individuals regardless of BMI. Furthermore, fewer than one-fourth of the participants were able to accurately estimate the caloric content of full meals—even among those with higher education levels. A trend toward underestimating caloric values was observed in individuals with excessive body weight. When assessing the feeling of fullness after meals, only two dishes (oatmeal and pasta) showed statistically significant differences. The lower satiety reported after consuming these meals may lead to increased portion sizes and, consequently, contribute to the development of obesity.

This study has several limitations. One was the length of the questionnaire. Participants were informed that evaluating one meal would take only about 30 s. Still, as we learned, the total time required for the questionnaire to be completed was perceived as excessive. While a detailed questionnaire collects valuable data, it requires sustained attention from respondents, which is only sometimes feasible. Another limitation could be the relatively small sample size (205 participants), possibly due to some patients discontinuing the completing of the questionnaire. Assuming the distribution of subgroups of 92, 66, and 47 people, assuming that the data are nonparametric, and a medium effect size, significance level of 0.05, the power of the test is 83% (which is over the standard 80%). However, when interpreting the results of our study, it should be taken into account that it was conducted on a specific number of respondents and the results require confirmation in a study on a larger population.

Additionally, the fact that participants self-reported their anthropometric data (height, weight) is also a limitation due to the need for the actual verification of this data. Incorrectly provided data by participants may cause errors in the study results. Education level was also biased towards participants with higher than lower levels of education. That bias may have occurred because the study group was part of a cohort interested in weight control and healthy nutrition issues active on various websites. Thus, individuals with a lower awareness and self-care levels were naturally underrepresented. The study was conducted in the Polish population. According to GUS data from 2021, 60% of the Polish population aged 25–44 had achieved a higher education level. With a study average age of 39.7 years, the high percentage of higher education respondents fits with general population data [32]. On the other hand, our findings highlight the challenge of food education: when well-educated individuals struggle to estimate meals, they need help accurately estimating meal sizes. The large number of meals assessed, covering a wide variety of foods consumed daily by people worldwide, and their careful visual presentation and description are among the greatest strengths of our study.

## 5. Conclusions

Patients’ BMI, in general, seems not to impact the perception of meal sizes, satiety sensation, and the conviction that the meal is healthy. The only two exceptions were the oatmeal and brown pasta dishes, which the obese respondents perceived as less satiating.

The majority of the population estimate the caloric value of their meals inappropriately.

There might be a relationship between BMI and caloric content estimation—patients who are overweight or obese are more likely to underestimate the caloric value of standard meals than ones with a normal BMI; however, the results were not statistically significant.

The incorrect estimation of meal caloric value may be the reason for increase in portion sizes and indirectly contribute to obesity development.

## Figures and Tables

**Figure 1 nutrients-17-02054-f001:**
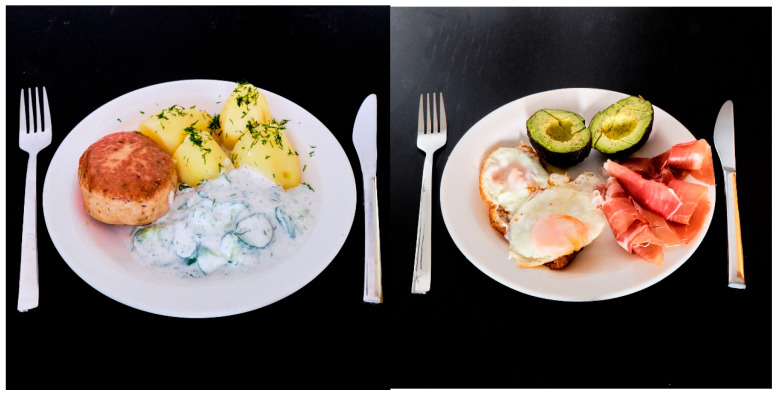
Examples of pictures used in questionnaire (**left**: minced meat cutlet with potatoes and cucumber salad; **right**: eggs with Parma ham and avocado).

**Table 1 nutrients-17-02054-t001:** The study group characteristics.

Variable		*n* [%] or Mean ± SD
Gender	Female	149 [72.7]
	Male	56 [27.3]
Age	(years)	39.7 ± 13.1
Weight	(kg)	78.2 ± 20.9
Height	(cm)	169.9 ± 8.6
BMI	(kg/m^2^)	27.0 ± 6.3
BMI (ranges)	<25	92 [44.9]
	25–29.9	66 [30.7]
	≥30	47 [24.4]
Education *n* [%]	Academic	142 [69.3]
	High School	62 [30.2]
	Primary Education	1 [0.5]

**Table 2 nutrients-17-02054-t002:** Breakfast meals.

Feature	Description	BMI			*p*
		<25 (*n* = 92)	25–29.9 (*n* = 66)	≥30 (*n* = 47)	
Whole grain bread sandwiches with cottage cheese, bell peppers, and tomatoes (352 kcal)					
Meal size [1–10]	Med. (IQR)	7 (2)	7 (2)	8 (3)	0.604
Fullness feeling [1–10]	Med. (IQR)	8 (2)	8 (2)	8 (3)	0.723
Estimated calorie value [kcal]	Med. (IQR)	400 (200)	400 (225)	350 (120)	0.628
“Do you think this meal is healthy?”	Y/N	91/1	64/2	46/1	0.818
“Will you feel full after eating this meal?”	Y/N	2/90	4/62	2/45	0.468
“Will you NOT feel hunger/craving in the next 2 h?”	Y/N	76/16	59/7	39/8	0.461
Fried eggs in olive oil with Parma ham and avocado (665 kcal)					
Meal size [1–10]	Med. (IQR)	7 (3)	7 (2)	7 (3)	0.997
Fullness feeling [1–10]	Med. (IQR)	8 (2)	8 (3)	8 (2)	0.979
Estimated calorie value [kcal]	Med. (IQR)	500 (403)	400 (300)	450 (300)	0.641
“Do you think this meal is healthy?”	Y/N	85/7	63/3	46/1	0.448
“Will you feel full after eating this meal?”	Y/N	4/88	2/64	3/44	0.690
“Will you NOT feel hunger/craving in the next 2 h?”	Y/N	79/13	57/9	42/5	0.839
Oatmeal with yogurt (519 kcal)					
Meal size [1–10]	Med. (IQR)	7 (2)	6 (3)	7 (3)	0.377
Fullness feeling [1–10]	Med. (IQR)	8 (3)	7 (2)	7 (4)	0.338
Estimated calorie value [kcal]	Med. (IQR)	350 (170)	300 (150)	350 (160)	0.119
“Do you think this meal is healthy?”	Y/N	87/5	64/2	44/3	0.768
“Will you feel full after eating this meal?”	Y/N	5/87	8/58	9/38	0.043
“Will you NOT feel hunger/craving in the next 2 h?”	Y/N	72/20	48/18	36/11	0.721

Y—yes, N—no, Med.—Mediana, IQR—interquartire rang.

**Table 3 nutrients-17-02054-t003:** Lunch/dinner meals.

Feature	Description	BMI			*p*
		<25 (*n* = 92)	25–29.9 (*n* = 66)	≥30 (*n* = 47)	
Hamburger with fries (851 kcal)					
Meal size [1–10]	Med. (IQR)	7 (3)	7 (3)	7 (2)	0.922
Fullness feeling [1–10]	Med. (IQR)	8 (3)	8 (3)	8 (3)	0.625
Estimated calorie value [kcal]	Med. (IQR)	1354 (204)	1541 (412)	1704 (37)	0.550
“Do you think this meal is healthy?”	Y/N	2/90	1/65	0/47	0.799
“Will you feel full after eating this meal?”	T/N	20/72	14/52	8/39	0.796
“Will you NOT feel hunger/craving in the next 2 h?”	T/N	59/33	42/24	31/16	0.966
Sushi (374 kcal)					
Meal size [1–10]	Med. (IQR)	7 (3)	6 (2)	6 (3)	0.712
Fullness feeling [1–10]	Med. (IQR)	7 (4)	6 (3)	7 (3)	0.409
Estimated calorie value [kcal]	Med. (IQR)	450 (250)	475 (300)	500 (150)	0.968
“Do you think this meal is healthy?”	Y/N	82/10	53/13	44/3	0.091
“Will you feel full after eating this meal?”	Y/N	19/73	14/52	5/42	0.283
“Will you NOT feel hunger/craving in the next 2 h?”	Y/N	66/26	41/25	34/13	0.365
Cutlet with potatoes and salad (*n* = 187) (476 kcal)					
Meal size [1–10]	Med. (IQR)	9 (1)	9 (3)	9 (2)	0.903
Fullness feeling [1–10]	Med. (IQR)	9 (2)	9 (2)	9 (2)	0.880
Estimated calorie value [kcal]	Med. (IQR)	650 (300)	600 (300)	600 (300)	0.845
“Do you think this meal is healthy?”	Y/N	56/27	40/20	32/12	0.780
“Will you feel full after eating this meal?”	Y/N	2/81	2/58	0/44	0.686
“Will you NOT feel hunger/craving in the next 2 h?”	Y/N	82/1	58/1	43/1	0.999
Greek salad (351 kcal)					
Meal size [1–10]	Med. (IQR)	8 (2)	8 (2)	8 (4)	0.783
Fullness feeling [1–10]	Med. (IQR)	8 (2)	8 (2)	8 (4)	0.392
Estimated calorie value [kcal]	Med. (IQR)	400 (285)	375 (150)	350 (165)	0.559
“Do you think this meal is healthy?”	Y/N	92/0	66/0	46/1	0.229
“Will you feel full after eating this meal?”	Y/N	8/84	4/62	7/40	0.286
“Will you NOT feel hunger/craving in the next 2 h?”	Y/N	70/22	47/19	31/16	0.441
Pizza with salami and pepperoni (1012 kcal)					
Meal size [1–10]	Med. (IQR)	10 (1)	10 (1)	10 (2)	0.943
Fullness feeling [1–10]	Med. (IQR)	10 (1)	10 (1)	10 (2)	0.499
Estimated calorie value [kcal]	Med. (IQR)	1000 (700)	1000 (500)	1000 (800)	0.918
“Do you think this meal is healthy?”	Y/N	5/87	4/62	6/41	0.311
“Will you feel full after eating this meal?”	Y/N	2/90	0/66	2/45	0.272
“Will you NOT feel hunger/craving in the next 2 h?”	Y/N	86/6	61/5	44/3	0.999
Brown pasta with pesto, tomatoes, and mozzarella (541 kcal)					
Meal size [1–10]	Med. (IQR)	8 (2)	8 (3)	8 (2)	0.023
Fullness feeling [1–10]	Med. (IQR)	9 (2)	9 (3)	8 (3)	0.036
Estimated calorie value [kcal]	Med. (IQR)	550 (300)	500 (265)	450 (200)	0.133
“Do you think this meal is healthy?”	Y/N	88/4	61/5	47/0	0.139
“Will you feel full after eating this meal?”	Y/N	3/89	1/65	1/46	0.853
“Will you NOT feel hunger/craving in the next 2 h?”	Y/N	87/5	62/4	44/3	0.999
Tomato cream soup (154 kcal)					
Meal size [1–10]	Med. (IQR)	6 (3)	6 (2)	6 (3)	0.815
Fullness feeling [1–10]	Med. (IQR)	6 (3)	5 (3)	6 (4)	0.293
Estimated calorie value [kcal]	Med. (IQR)	300 (200)	250 (200)	300 (150)	0.224
“Do you think this meal is healthy?”	Y/N	89/3	65/1	47/0	0.571
“Will you feel full after eating this meal?”	Y/N	19/73	18/48	13/34	0.531
“Will you NOT feel hunger/craving in the next 2 h?”	Y/N	37/55	26/40	26/21	0.171

Y—yes, N—no, Med.—median, IQR—interquartile range.

**Table 4 nutrients-17-02054-t004:** Real and estimated caloric value of the meals as given by all participants (*n* = 205). Only the energy value of porridge with yogurt and eggs with avocado (breakfast meals) was underestimated on average, while the caloric content of all other meals was overestimated.

	Meal	Exact Caloric Value	Estimated Caloric Parameters			%pts Correctly Estimating *
			M	SD	CI (0.95)	
1	Oatmeal with yogurt	519	374	225	100–900	8.8
2	Greek salad	351	444	307	100–1500	9.3
3	Whole grain bread sandwiches with cottage cheese, bell peppers, and tomatoes	352	446	371	150–1500	9.8
4	Tomato cream soup	154	322	278	80–900	9.8
5	Cutlet with potatoes and salad	476	764	627	280–2000	14.1
6	Fried eggs in olive oil with Parma ham and avocado	665	498	290	120–1200	17.6
7	Hamburger with fries	851	1086	681	459–3000	20.5
8	Pizza with salami and pepperoni	1012	1146	852	300–2500	21.9
9	Sushi	374	536	465	150–1500	22.4
10	Brown pasta with pesto, tomatoes, and mozzarella	541	583	400	150–1500	21.0

M—mean, SD—standard deviation, CI—confidence interval, * Percentage of patients who accurately (90–110% exact caloric value) estimated caloric value of meals.

**Table 5 nutrients-17-02054-t005:** Odds ratio for overestimation of the meals’ calorie value in individuals with BMI > 25 kg/m^2^. Only the caloric content of pasta was found to be statistically significantly more often underestimated.

Meal	OR	CI (0.95)	*p*
Hamburger with fries	0.67	0.38–1.16	0.162
Whole grain bread sandwiches with cottage cheese, bell peppers, and tomatoes	0.63	0.35–1.1	0.123
Sushi	0.8	0.44–1.46	0.540
Cutlet with potatoes and salad	1.03	0.51–2.08	1.00
Greek salad	0.7	0.4–1.22	0.261
Pizza with salami and pepperoni	0.83	0.46–1.46	0.559
Brown pasta with pesto, tomatoes, and mozzarella	0.51	0.29–0.89	0.022
Tomato cream soup	0.74	0.38–1.44	0.404
Fried eggs in olive oil with Parma ham and avocado	0.57	0.29–1.13	0.122
Oatmeal with yogurt	0.9	0.35–2.31	0.804

OR—odds ratio, CI—confidence interval, *p*—*p*-value.

## Data Availability

The original contributions presented in the study are included in the article/Appendix A, further inquiries can be directed to the corresponding author.

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
