# Peer review of "Analysis of Food Perception in Slim, Overweight, or Obese Individuals"

_nutrients, 2025, doi:10.3390/nu17132054_

Round 1
Reviewer 1 Report
Comments and Suggestions for Authors
Dear Authors,
I have reviewed your article with the Title: “Analysis of Food Perception in Slim, Overweight or Obese Individuals“ and present my observations:
The study design is a cross-sectional, questionnaire-based study, which is appropriate for the stated objective of assessing portion perception and caloric estimation. However, there are several important limitations that should be addressed:
- The authors report that no validated questionnaire suited their needs, and thus they developed one based solely on their professional experience. While this approach is understandable, it would have been essential to detail the development and validation process of the new instrument, or at least to justify its reliability through preliminary testing or expert review.
- The overarching topic of portion perception and caloric estimation has been previously explored in the literature. However, the manuscript does not clearly state how this study differs from existing work. The lack of a comprehensive literature review in the introduction further limits the ability to assess the study’s originality and the specific knowledge gap it addresses.
- While the total sample size (205 participants) is reasonable, the unequal distribution between BMI categories (92 normal weight, 63 overweight, and 50 obese) may reduce the statistical power for between-group comparisons. A more balanced distribution—ideally with approximately 100 participants per group (1:1:1 ratio)—would have been preferable, provided it was supported by an a priori power analysis.This imbalance increases the risk of Type II errors (false negatives), potentially masking true differences between groups.
- Anthropometric data (weight, height, BMI) were self-reported, which may introduce substantial bias. Prior studies have demonstrated that individuals tend to underestimate their weight and overestimate their height, affecting BMI accuracy. This limitation should be explicitly acknowledged and discussed in the manuscript.
- Formatting inconsistencies are present in the tables, including extra spaces and misaligned elements, which negatively affect readability.
- Some sentences—particularly in the methodology section—are overly long and complex, reducing clarity. For instance, the questionnaire description spans multiple sentences without clear segmentation or logical flow.
- The conclusions section requires reformulation to reflect the actual findings more accurately and cautiously.
- There are inconsistencies in the use of verb tenses throughout the methodology and results sections. For example, section 2.1 includes 'The questionnaire was shared' (past tense), followed by 'Each adult person who accessed these websites is welcome' (present tense), which disrupts narrative coherence.
- There are occasional grammatical issues, such as the sentence in the abstract: 'It can lead to consuming more significant portions of meals in people with overweight and obesity of meals in people who are overweight and obese,' which contains an unnecessary and confusing repetition.
Author Response
Dear Reviewer 1,
First of all, we would kindly like to thank you for the review. We have corrected the manuscript according to all your valuable comments and suggestions. We hope the corrected manuscript will meet your all expectations. Below, we have attached the answers for all of your questions and suggestions.
All changes and updates were marked red in main the text.
The study design is a cross-sectional, questionnaire-based study, which is appropriate for the stated objective of assessing portion perception and caloric estimation. However, there are several important limitations that should be addressed:
- The authors report that no validated questionnaire suited their needs, and thus they developed one based solely on their professional experience. While this approach is understandable, it would have been essential to detail the development and validation process of the new instrument, or at least to justify its reliability through preliminary testing or expert review.
We have to agree that the questionnaire was no validated, however we did not find suitable questionnaire for the hypothesis and study design. The senior author of the manuscript is national expert in diabetology and obesitology, and he approved study design and the questionnaire.
- The overarching topic of portion perception and caloric estimation has been previously explored in the literature. However, the manuscript does not clearly state how this study differs from existing work. The lack of a comprehensive literature review in the introduction further limits the ability to assess the study’s originality and the specific knowledge gap it addresses.
In our clinical observation we have noticed some patterns that majority of patients declare, regarding their nutrition style and diet. We design the study to confirm this observation, however the statistical analysis did not fully confirmed the hypothesis. There were some previous studies, presented in discussion, nevertheless, they were unclear and ambiguous. Moreover, in the authors’ country, which has one of the highest percentages of obese or overweight people in Europe, such studies were not previously performed.
- While the total sample size (205 participants) is reasonable, the unequal distribution between BMI categories (92 normal weight, 63 overweight, and 50 obese) may reduce the statistical power for between-group comparisons. A more balanced distribution—ideally with approximately 100 participants per group (1:1:1 ratio)—would have been preferable, provided it was supported by an a priori power analysis. This imbalance increases the risk of Type II errors (false negatives), potentially masking true differences between groups.
The design of the study was to obtain such (1:1:1) ratios, with the study group >200 participants. However, we did not reach enough volunteers, and due to lack of new respondets we performed final calculation on the study group presented in „Materials and methods”. We completely confirm that this kind of bias could increase the risk of errors, nevertheless, a comparsion of patients with „normal” and „excessive” BMI (BMI>25), which has ratio of 92:113, did also not bring any statistical significances.
- Anthropometric data (weight, height, BMI) were self-reported, which may introduce substantial bias. Prior studies have demonstrated that individuals tend to underestimate their weight and overestimate their height, affecting BMI accuracy. This limitation should be explicitly acknowledged and discussed in the manuscript.
Text updated
- Formatting inconsistencies are present in the tables, including extra spaces and misaligned elements, which negatively affect readability.
Tables updated
- Some sentences—particularly in the methodology section—are overly long and complex, reducing clarity. For instance, the questionnaire description spans multiple sentences without clear segmentation or logical flow.
Text corrected.
- The conclusions section requires reformulation to reflect the actual findings more accurately and cautiously.
Text corrected.
- There are inconsistencies in the use of verb tenses throughout the methodology and results sections. For example, section 2.1 includes 'The questionnaire was shared' (past tense), followed by 'Each adult person who accessed these websites is welcome' (present tense), which disrupts narrative coherence.
Tenses incoherence updated.
- There are occasional grammatical issues, such as the sentence in the abstract: 'It can lead to consuming more significant portions of meals in people with overweight and obesity of meals in people who are overweight and obese,' which contains an unnecessary and confusing repetition.
Error corrected.

Reviewer 2 Report
Comments and Suggestions for Authors
Thank you for allowing me to review the article "Analysis of Food Perception in Slim, Overweight or Obese Individuals." I have a few questions and comments:
-
In the abstract, the authors emphasize that “Nevertheless, patients with excessive body weight (overweight and obesity) are more likely to underestimate the caloric value of full meals than people with average body weight.” Which data in the paper support this conclusion? In the results section of the abstract, the authors only describe some non-significant results.
-
Regarding the 205 volunteers—where were they recruited from? Were they hospital patients, or were they recruited through advertisements?
-
In the results section, the authors state: “Individuals with excessive body weight (BMI ≥ 25kg/m²) had a higher risk of underestimating meal calories compared to those with normal body weight, except for the meal - cutlet with potatoes and salad, where the risk of overestimating the meal was minimally higher [OR 1.03 (CI 0.95, 0.51–2.08), p = 0.559].” Where exactly is the data supporting this statement? Which part of the data supports the claim of a “higher risk of underestimating”?
-
Given the large number of comparisons and results reported, did the authors account for multiple comparison correction?
Author Response
Dear Reviewer 2
First of all, we would kindly like to thank you for the review. We have corrected the manuscript according to all your valuable comments and suggestions. We hope the corrected manuscript will meet your all expectations. Below, we have attached the answers for all of your questions and suggestions.
All changes and updates were marked blue in the text.
- In the abstract, the authors emphasize that “Nevertheless, patients with excessive body weight (overweight and obesity) are more likely to underestimate the caloric value of full meals than people with average body weight.” Which data in the paper support this conclusion? In the results section of the abstract, the authors only describe some non-significant results.
The abstract is updated. There is tendency to underestiamte caloric valure by people with excesive body weight but it is not statistically significant.
2. Regarding the 205 volunteers—where were they recruited from? Were they hospital patients, or were they recruited through advertisements?
As it was mentioned in the part „Materials and methods” : The anonymous questionnaire was shared and made available on several popular websites devoted to healthy nutrition and weight management or on a social media. Each adult person who accessed these websites was welcome to fill it in. We have also encouraged patients in outpatient clinics to fill out the forms. Volunteers were recruted by the Internet or in outpatient clinics (like General Practice Cinic, Dietetic Clinic, Diabetologist clinic). There was only one exception – participant have to by older than 18 years old.
3. In the results section, the authors state: “Individuals with excessive body weight (BMI ≥ 25kg/m²) had a higher risk of underestimating meal calories compared to those with normal body weight, except for the meal - cutlet with potatoes and salad, where the risk of overestimating the meal was minimally higher [OR 1.03 (CI 0.95, 0.51–2.08), p = 0.559].” Where exactly is the data supporting this statement? Which part of the data supports the claim of a “higher risk of underestimating”?
Text updated. There is tendecy to underestimate meal calories by people with overweight or obesity but it is not statistically significant.
4. Given the large number of comparisons and results reported, did the authors account for multiple comparison correction?
For nominal data chi-square and Fisher exact tests were performed. For quantitative data Kruskall-Wallis was performed, followed by post-hoc tests (Bonferroni).

Round 2
Reviewer 1 Report
Comments and Suggestions for Authors
Dear Authors,
I appreciate your efforts to address the previous recommendations and the modifications you've made to the manuscript. While these changes show your commitment to improving the work, additional enhancements would help meet Nutrients' publication standards:
Briefly describe the expert-guided development of your questionnaire
Expand the introduction to highlight how your study fills specific gaps in the literature
Emphasize the photographic methodology with standardized cutlery as a methodological strength
Add a brief power analysis discussion and acknowledge sample size limitations
Your research addresses an important topic in nutrition and obesity management, and these targeted improvements would enhance its impact and publication potential.
Author Response
Dear Reviewer 1, We would like again to thank You for Your comments and time spent on the manuscript evaluation. Thank you again for giving us the oportunity to revise and improve our article. All Your suggestions were considered and included in the manuscript text with red colour.
- Briefly describe the expert-guided development of your questionnaire
We added a paragraph in „Materials and Methods” describing development of questionare.
- Expand the introduction to highlight how your study fills specific gaps in the literaturÄ™
Introduction updated
- Emphasize the photographic methodology with standardized cutlery as a methodological strength
We added a paragraph in „Materials and Methods” emphasizinig the photographic methodology. - Add a brief power analysis discussion and acknowledge sample size limitations
We added a paragraph in „Discussion” regarding limitations of statistics.

Reviewer 2 Report
Comments and Suggestions for Authors
I am not satisfied with the author's response. I believe the author did not genuinely attempt to address my comments.
Author Response
Dear Reviewer 2,
First of all we would like to sincerely apologize for the unsatisfactory reply to the review. This situation resulted from an incomplete understanding of the suggestions you indicated.
We would like to address all of your previous remarks in a more holistic way. We hope that revised text and the answers themself will be satisfactory this time. We wish the improvement of our manuscript will be suitable, and allow the article to be published in Nutrients.
All your suggestions are marked in blue colour.
- In the abstract, the authors emphasize that “Nevertheless, patients with excessive body weight (overweight and obesity) are more likely to underestimate the caloric value of full meals than people with average body weight.” Which data in the paper support this conclusion? In the results section of the abstract, the authors only describe some non-significant results.
Dear reviewer, the abstract was rewritten due to your valuable comment. It is now more coherent and clear and it reflects the results obtained in our study. We calculated the risk of overestimating the caloric value of meals by people with BMI > 25 kg/m2 (OR - odds ratio). Most of the results are not statistically significant (with one exception - pasta meal). The results are presented in Table 5.
2. Regarding the 205 volunteers—where were they recruited from? Were they hospital patients, or were they recruited through advertisements?
The anonymous questionnaire was shared and made available on several popular websites devoted to healthy nutrition and weight management or on social media. We have also encouraged patients in outpatient clinics to fill out the forms. Volunteers were recruited by the Internet or in outpatient clinics (like General Practice Clinic, Dietetic Clinic, Diabetologist Clinic. We detaily described recruitment in „Materials and methods” .
3. In the results section, the authors state: “Individuals with excessive body weight (BMI ≥ 25kg/m²) had a higher risk of underestimating meal calories compared to those with normal body weight, except for the meal - cutlet with potatoes and salad, where the risk of overestimating the meal was minimally higher [OR 1.03 (CI 0.95, 0.51–2.08), p = 0.559].” Where exactly is the data supporting this statement? Which part of the data supports the claim of a “higher risk of underestimating”?
There was only a tendency to underestimate meal calories by patients with overweight or obesity but it was not statistically significant. We corrected the article text to be clear and more understandable.
4. Given the large number of comparisons and results reported, did the author's account for multiple comparison corrections?
For nominal data chi-square and Fisher exact tests were performed. For quantitative data Kruskall-Wallis was performed, followed by post-hoc tests (Bonferroni). All of those tests were performed to increase the accuracy of the calculations and in order to validate the results. We have also added a paragraph in the discussion describing limitations correlated with statistical analysis.

Round 3
Reviewer 1 Report
Comments and Suggestions for Authors
Dear Authors,
Thank you for submitting your revised manuscript "Analysis of food perception in lean, overweight and obese individuals" to Nutrients. I very much appreciate your efforts to improve this material, I am pleased to inform you that your paper better includes the following two sentences regarding the limitations of the study. I have suggested wording , but please adapt them as you consider appropriate:
1.
"A limitation of our study is the absence of a formal power analysis to determine the optimal sample size. The unequal distribution between BMI groups (92 normal-weight participants, 63 overweight participants, and 47 obese participants) may have reduced the statistical power to detect significant differences, especially for the obese group which had the smallest sample size. This imbalance could have increased the risk of type II (false negative) errors, meaning that it is possible that some real differences between groups were not detected."
2.
"Another limitation is our reliance on self-reported anthropometric data for BMI calculations. Previous research has shown that individuals tend to underestimate their weight and overestimate their height in self-reports, which may lead to underestimation of BMI values. This potential measurement bias could affect the accuracy of participant classification into BMI categories and subsequently influence the interpretation of our results.”